# Different roles of concurring climate and regional land-use changes in past 40 years' insect trends

Felix Neff [1] ✉, Fränzi Korner-Nievergelt [2], Emmanuel Rey [3], Matthias Albrecht[1], Kurt Bollmann [4], Fabian Cahenzli [5], Yannick Chittaro [3], Martin M. Gossner [6,7], Carlos Martínez-Núñez [1], Eliane S. Meier [1], Christian Monnerat[3], Marco Moretti [4], Tobias Roth[8,9], Felix Herzog [1] & Eva Knop[1,10]

Climate and land-use changes are main drivers of insect declines, but their combined effects have not yet been quantified over large spatiotemporal scales. We analysed changes in the distribution (mean occupancy of squares) of 390 insect species (butterflies, grasshoppers, dragonflies), using 1.45 million records from across bioclimatic gradients of Switzerland between 1980 and 2020. We found no overall decline, but strong increases and decreases in the distributions of different species. For species that showed strongest increases (25% quantile), the average proportion of occupied squares increased in 40 years by 0.128 (95% credible interval: 0.123–0.132), which equals an average increase in mean occupancy of 71.3% (95% CI: 67.4–75.1%) relative to their 40-year mean occupancy. For species that showed strongest declines (25% quantile), the average proportion decreased by 0.0660 (95% CI: 0.0613–0.0709), equalling an average decrease in mean occupancy of 58.3% (95% CI: 52.2–64.4%). Decreases were strongest for narrow-ranged, specialised, and cold-adapted species. Short-term distribution changes were associated to both climate changes and regional land-use changes. Moreover, interactive effects between climate and regional land-use changes confirm that the various drivers of global change can have even greater impacts on biodiversity in combination than alone. In contrast, 40-year distribution changes were not clearly related to regional land-use changes, potentially reflecting mixed changes in local land use after 1980. Climate warming however was strongly linked to 40-year changes, indicating its key role in driving insect trends of temperate regions in recent decades.

Being the most diverse group of animals[1,2], insects represent a major part of Earth's biodiversity and contribute to essential ecosystem services such as pollination and pest control[3,4]. Therefore, recent reports on their decline[5–7] raised major concerns in the scientific community[8] as well as among policymakers, stakeholders and the general public. Yet, the generality of insect decline across regions,

ecosystems and insect groups remains a matter of debate[9,10]. Also, we are only starting to understand what the main drivers of the observed trends in insect populations are[7]. To date, climate change and land-use change, including the intensification of agricultural practices such as pesticide use, are considered important drivers[6,11–13]. As climate and land-use changes are expected to further increase their influence in the

coming decades in many regions of the world[14], their combined impacts on insect population trends are of particular interest[15].

Recent spatial analyses suggest that insect communities may be shaped by both additive and interactive effects of climate, land use and changes therein[16–18]. Such analyses based on space-for-time substitution provide valuable insights into insect declines in the absence of time-series data[19], but to understand past developments of insect populations, time series are key[20]. In particular, for linking the combined effects of climate and land-use changes to insect trends over relevant temporal scales (i.e., decades[19]), data on insects in combination with data on these drivers over large temporal and spatial scales is crucial, but rarely available. The few studies that simultaneously addressed climate and land-use effects on insect trends highlight the importance of both drivers for observed changes[11,21–23]. However, they mostly lack quantification of additive and interactive effects and are all based on single taxonomic groups or subgroups with restricted ecological characteristics, strongly limiting the generality of their findings[24].

Here, we analysed trends in species distributions from records of three diverse and widespread insect groups with different ecological characteristics—butterflies (refers here to all Papilionoidea as well as to Zygaenidae moths), grasshoppers (refers here to all Orthoptera) and dragonflies (refers here to all Odonata)—in Switzerland across a 40-year period (1980–2020). All three groups include species ranging from habitat specialists to generalists as well as from cold- to warm-adapted (Fig. 1a and Supplementary Table S1). We quantified the main and interactive effects of climate and land-use changes at a regional scale on distribution trends by assessing trends separately for nine bioclimatic zones (~1000–10,000 km²), which were defined by biogeographic regions and elevation (Fig. 1b). These zones experienced different trajectories of climate change (annual mean temperature, temperature seasonality, summer precipitation) and land-use change (total agricultural area, grassland-use intensity, crop-use intensity) across the study period (Fig. 1c). In addition to using independent distribution trends for the nine zones, we assessed trends for consecutive 5-year intervals. The aim of the choice of this spatial and temporal resolution was to get a sufficient number of replicates of trends under different climate and land-use conditions that showed enough variance while their covariance was low (Supplementary Fig. S1). Also, we considered the 5-year intervals to be most meaningful for the studied insect groups from an ecological perspective. Nonetheless, we run the analyses based on 10-year intervals and confirmed the main findings. We further analysed how trends in species dis-

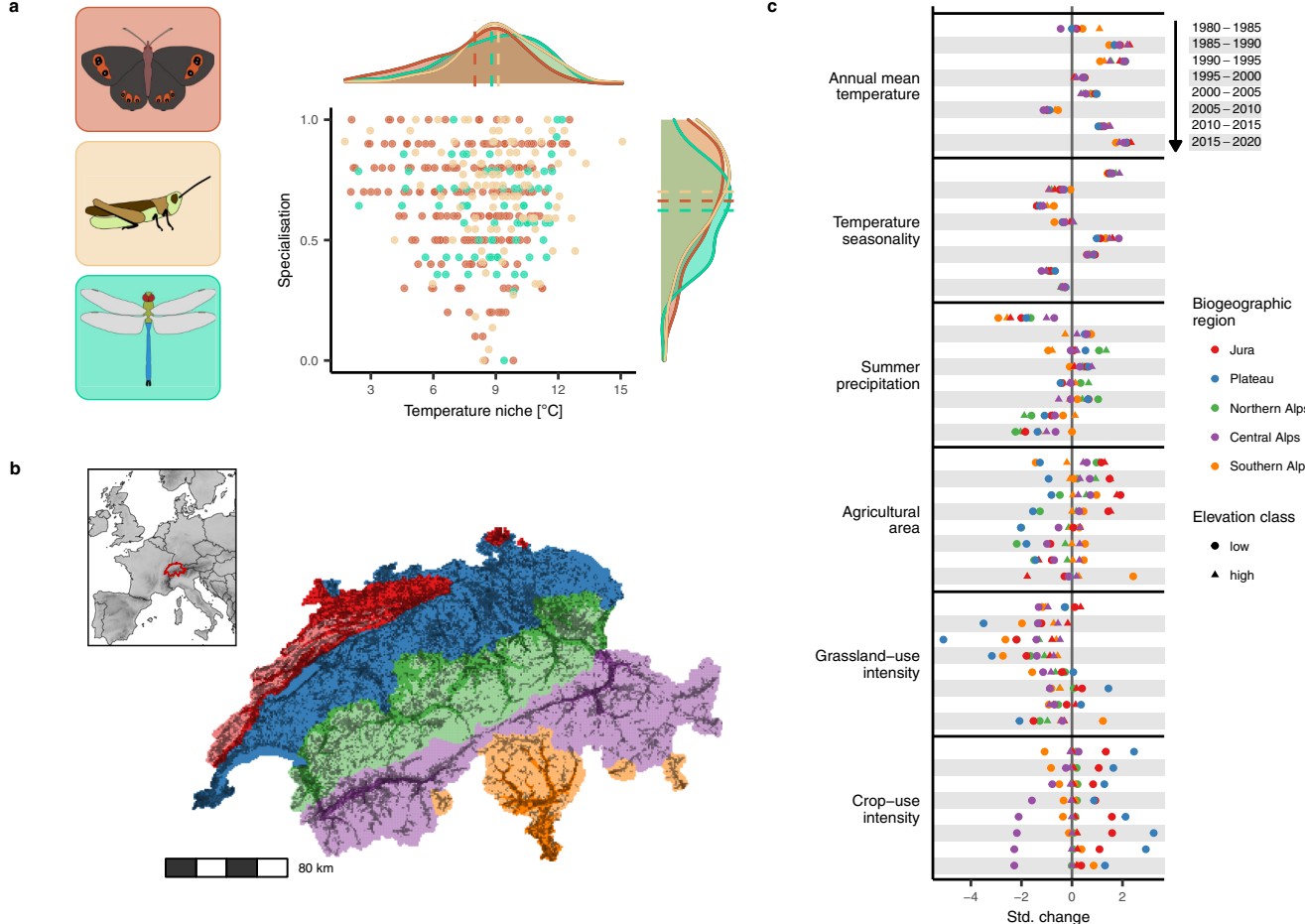

**Fig. 1 | Study species, study region, and climate and land-use changes. a** In total, 215 butterfly species, 103 grasshopper species and 72 dragonfly species were analysed, which covered a gradient in habitat specialisation (0: lowest specialisation; 1: highest specialisation; based on literature-derived habitat preferences) and in preferred temperature niches (average annual temperatures of Europe-wide distribution). Curves show marginal density distributions per group and trait; dashed lines indicate means. **b** Switzerland, the study country situated along the Alps in Central Europe, was divided into five biogeographic regions indicated by colours (legend in (**c**)) and two elevation classes (above and below 1000 m asl.;

high elevation not distinguished for the low-elevational Plateau region) indicated by shadings (strong colour for low elevation), resulting in nine bioclimatic zones. Dark squares in the map show squares for which data of at least one insect group were analysed (darker colours indicate more records, Supplementary Fig. S2a). **c** Changes of six potential driver variables were assessed for eight focal 5-year intervals in the study period (arranged from top to bottom for each variable) for the nine bioclimatic zones (combinations of biogeographic region and elevation class). For all variables, change was standardised to standard deviation 1.

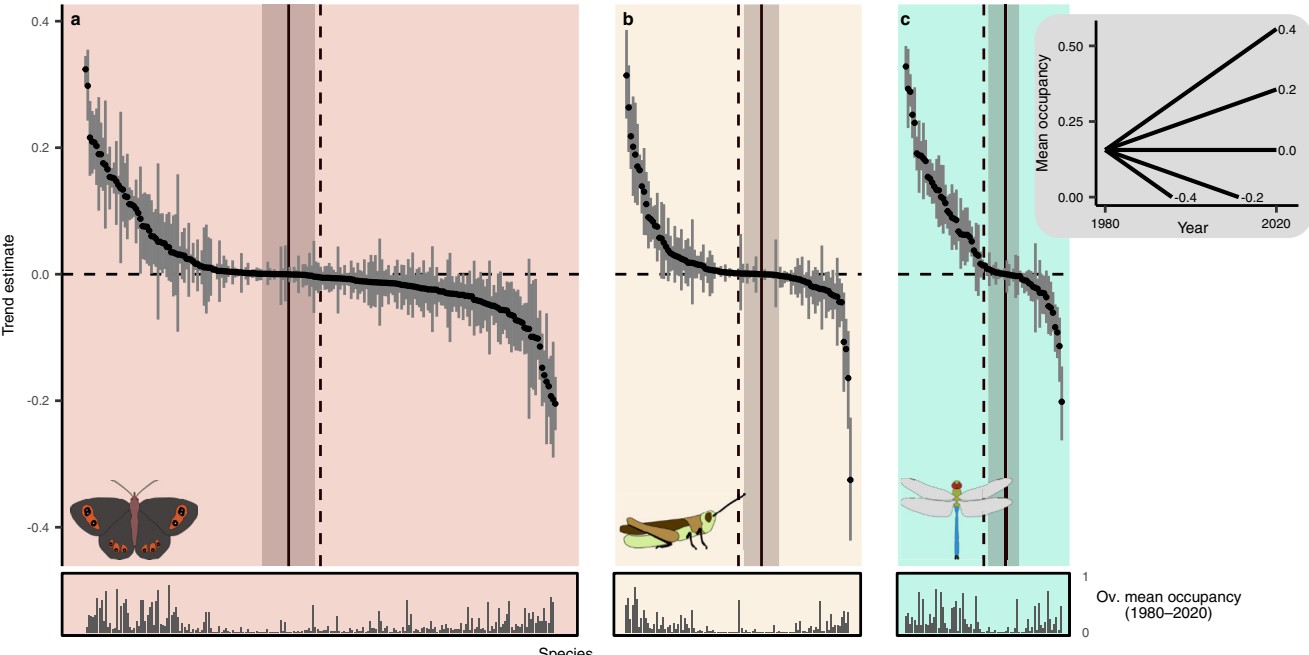

**Fig. 2 | Trend estimates of mean occupancy across 40 years (1980–2020).** 40-year trend estimates are shown for the (**a**) 215 butterfly species, the (**b**) 103 grasshopper species and the (**c**) 72 dragonfly species. Species are ordered along the point estimate of their trend (mean of posterior distribution), vertical segments show the 95% credible intervals. Trend estimates reflect a 40-year change in mean occupancy, which is illustrated in the inset plot at the right starting from the overall mean occupancy (mean across all species and years). The vertical dashed lines show the median of the number of species, the vertical solid lines show where negative trend estimates change to positive trend estimates along with a bootstrap 95% confidence interval (*n* = 9999). Bars in the bottom panels show 40-year mean occupancy estimates for the whole of Switzerland for each species.

tributions relate to species traits (temperature niche, habitat specialisation) and expected that responses to climate change differed among species with different temperature niches, whereas responses to land-use change differed among differently specialised species.

We included 1,448,134 records (879,207 butterflies, 272,863 grasshoppers, 296,064 dragonflies) (Supplementary Fig. S2) from a curated database that collates data from various projects, experts and naturalists. We determined annual mean occupancy of 1 × 1 km squares in each of the nine bioclimatic zones for each of the 390 studied species (215 butterflies, 103 grasshoppers, 72 dragonflies; Supplementary Table S1) as a measure of species distribution. To account for differences in sampling scheme and effort, we used occupancy-detection models, which allow to correct for observer bias. Such models have repeatedly been applied to reconstruct long-term time series of insect distributions[25–27]. For butterflies, data from standardised 17-year monitoring were available, which showed trends that broadly aligned with the estimated mean occupancy trends, supporting the validity of our models (Supplementary Fig. S3). We found no general trend in insect distributions, but both increases and decreases of single species, with decreases being strongest for narrow-ranged, specialised, and cold-adapted species. While short-term trends were linked to climate change, regional land-use change and their interaction, long-term trends were best explained by climate warming, showing that accounting for climate change is key to understand changes of insect distributions in the last decades.

## Results and discussion
### Distribution trends across 40 years
Across the whole of Switzerland, there was no overall trend in species distributions, but strong decreases and increases of individual species (Fig. 2). In total, 203 species (95% CI: 193–215, 48.7–54.6% of species) showed a positive trend in mean occupancy between 1980 and 2020, and 187 species (95% CI: 176–198) showed a negative trend. For the 25%

species with the strongest negative trends (declining quarter), mean occupancy, i.e., the proportion of total squares that are occupied, on average decreased across the 40 years by 0.0660 (95% CI: 0.0613–0.0709), whereas for the 25% with the strongest positive trends (increasing quarter), mean occupancy increased by 0.128 (95% CI: 0.123–0.132). This represents an average increase of occupied squares of 71.3% (95% CI: 67.4–75.1%) relative to their 40-year mean occupancy for species of the increasing quarter, but an average decrease of 58.3% (95% CI: 52.2–64.4%) for species of the decreasing quarter (percentage changes based on linear model predictions of species' mean occupancy). These changes are considerable and reflect ongoing shifts in community composition. If declines continue at the same extent, they might ultimately lead to further local species extinctions (Supplementary Fig. S4), exacerbating the impoverishment of insect communities.

The ratio of species with positive and negative trends differed among the studied insect groups. While a majority of 120 butterfly species (95% CI: 112–129; 52.1–60.0%) showed a negative 40-year trend, majorities of 61 grasshopper species (95% CI: 55–67; 53.4–65.0%) and of 47 dragonfly species (95% CI: 44–52; 61.1–72.2%) showed positive trends. The predominant decrease in butterfly distributions aligns with reports on declines from other regions in Europe[27–30] and from other continents[11]. Also, for dragonflies, which showed largely increasing species trends, our results are similar to findings from other regions in Europe[26,27,31,32]. The majority of increasing trends in grasshoppers was smaller, which adds to previous studies showing tendencies for increasing trends[33–35] but also for decreasing trends[27]. Because dragonflies have an aquatic life stage, the high proportion of dragonflies species showing positive trends might reflect improvements of water quality and wetland habitats made in the last decades[36]. Generally, positive trends in aquatic insects have been suggested to be linked to such improvements[10,25]. Furthermore, the fact that grasshoppers and dragonflies, the two groups with a majority of increasing species,

contain species that were on average more warm-adapted than butterfly species (Fig. 1a) indicates the potential role of climate change in driving species trends, as has previously been suggested for increasing trends of grasshoppers[34] and dragonflies[26,27,32], and decreasing trends of butterflies[37], respectively. Our findings show that different taxonomic groups with distinct ecological characteristics show different temporal trends, calling for a multi-taxa approach in order to understand the generality of insect trends[24]. Furthermore, different trends across groups will result in changes of species interactions and consequently ecosystem functioning.

Forty-year mean occupancy was higher for species with an increasing trend (0.181, 95% CI: 0.172–0.191) than for species with a decreasing trend (0.128, 95% CI: 0.119–0.138; difference 0.0531, 95% CI: 0.0336–0.0711) (Fig. 2), showing that the distribution of more widespread species increased even more, whereas the distribution of narrow-ranged species became even more confined[31,38]. This finding also holds when considering the global distribution of species (Supplementary Fig. S5). The increases in widespread species mean that communities are getting more similar, resulting in the homogenisation of community composition[39], which is threatening the magnitude and stability of ecosystem service provision at the landscape scale[40].

### The role of climate and land-use changes

We used a regression model to link 5-year (short-term) species trends to changes in climate and regional land-use changes as well as to species traits (Fig. 3a). These short-term species trends were most strongly linked to variables of climate change (annual mean temperature, temperature seasonality, summer precipitation) and to a lesser degree also to regional land-use change (agricultural area, grassland-use intensity), as well as to the interactions between climate and land-use change (Fig. 3b and Supplementary Tables S2–S4). The predominant role of climate change was even clearer when we used the model estimates based on 5-year trends to quantify how well climate and land-use change trajectories explain the observed 40-year (long-term) species trends. To this end, we defined scenarios in which change of all climate and land-use variables across the 40 years was assumed to be zero (no change) and only single variables or combinations of two variables were kept at their original trajectories. Based on the parameter estimates from the regression model (Fig. 3a), we predicted 40-year species trends for these different scenarios and aligned them with the observed 40-year species trends. Clearly, variance in long-term trends was best explained by scenarios accounting for the observed annual mean temperature trajectory, i.e., for climate warming (Fig. 4 and Supplementary Fig. S6), showing the crucial role of climate change in shaping insect communities at a regional scale[17]. As such, we provide strong evidence that in recent decades, climate changes have replaced regional land-use changes as the main driving force of large-scale insect distribution changes in Switzerland, a finding that most likely also hold for other (temperate) regions[41].

The weak relation of species trends to regional land-use change, particularly at the long-term (Fig. 4), potentially reflects the fact that most of the region-wide detrimental land-use changes, such as systematic fertilisation of dry meadows or draining of wetlands, have occurred in Switzerland more than 40 years ago[42], as is also true for other regions in Europe[43,44]. Recent land-use changes included both, further intensification of agricultural practices at some sites, but also changes aimed at promoting biodiversity, such as the extensification and restoration of grasslands, at other sites. Overall, this resulted in mixed local land-use trends across spatiotemporal scales (Supplementary Fig. S7), resulting in a weaker link of insect trends with regional land-use changes as compared to climate changes. Yet, at the local scale (e.g., on a given patch), land-use change and intensification are known to be important drivers of local insect diversity[17] with consequences for insect population trends[6], which, when resulting in a large-scale increase of land-use intensity, clearly negatively affect biodiversity at

regional scales[45]. Thus, land-use changes, which might have been locally considerable (Supplementary Fig. S7), most probably contributed to changes in the observed insect communities in the last 40 years, but we could not detect such relations given the regional scale of our approach. Nevertheless, declines and increases of insect populations and their relations to drivers at the studied regional scales are most relevant to understand the status of biodiversity at large scales.

The relations between drivers and species trends at 5-year intervals apparent from the regression model offer insights into what drove short-term changes in insect distributions and how distributions might change for different future trajectories of these drivers (Fig. 3b and Supplementary Tables S2–S4). On average, short-term species trends across all insect groups were more negative in intervals of rapid warming (increasing annual mean temperature) and increasing temperature seasonality, both of which can result in non-optimal conditions for many species during at least parts of their life cycle[41]. That the negative relation to climate warming was less evident at high elevation might reflect ongoing range shifts[46] with species adapted to low elevations increasing in occupancy and high-elevation species prevailing and shifting to even higher elevations. Also, intervals of increasing summer precipitation were associated with more negative trends. Wetter weather conditions during the periods of highest activity could hamper population sizes through, e.g., increased larval mortality[47,48]. Relations to land-use variables were complex (Fig. 3b). On the one hand, short-term species trends at high elevation were positively linked to decreases in the agricultural area. At this elevation, decreases in the cover of agricultural land were related to increases in the cover of scrubs and forests (Supplementary Fig. S8). These woody habitats might, particularly at early stages of encroachment, provide valuable structures and suitable microclimatic conditions also for open-habitat species. On the other hand, grassland-use intensification tended to be negatively linked to short-term insect trends, again only at high elevation. This might be because high elevations still harbour many low-intensity grasslands (Supplementary Fig. S9), whose communities are still susceptible to land-use intensification. Maintaining and promoting low-intensity mountain grasslands, where also many of the species particularly susceptible to climate warming are prevalent, is thus key to prevent further species extinctions.

Grassland-use intensification was particularly negatively linked to short-term insect trends in periods of decreasing annual mean temperature, temperature seasonality, and summer precipitation (Fig. 3b). While the causal relations behind these and other interactions remain unclear given the correlative nature of this study, the interactive effects of climate and land-use changes on short-term trends indicate the complexity of climate and land-use interactions, and that they urgently require more attention. Particularly, there is need to consider such interactions when extrapolating to long-term insect trends in other regions and in future scenarios.

The important role of climate change in explaining species trends is supported by the fact that warm-adapted species tended to show positive trends, whereas cold-adapted species tended to show negative trends (Figs. 3b and 5), supporting the findings from previous studies[26,27,31,34,49]. Furthermore, these patterns seemed to be stronger in intervals of more rapidly increasing temperatures, although this interaction effect was weak (Fig. 3b). Similarly, trends of habitat specialists were more negative than those of habitat generalists (Figs. 3b and 5), which is in line with previous studies showing particularly strong declines in insect species specialised on certain habitats or food plants[23,27,31,34]. In summary, more cold-adapted species with restricted habitat ranges seem to be particularly threatened from ongoing global change, which in the case of Switzerland includes many alpine species. With temperature increase ongoing in the current century, many of these species will be further threatened. As such, we provide clear evidence that climate change is a key driver of insect trends in recent decades, which will have to be considered in future actions to conserve insect diversity.

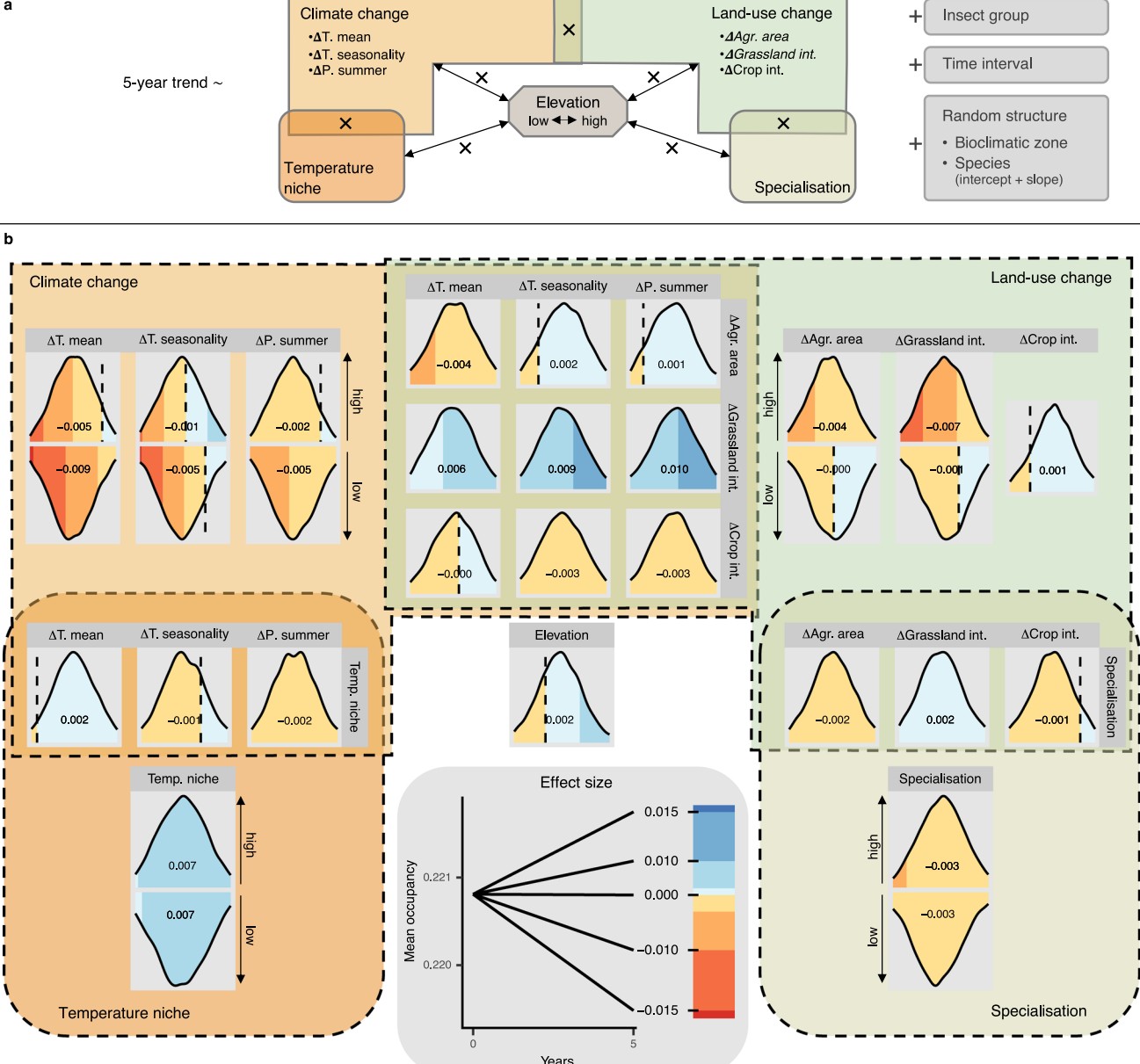

**Fig. 3 | Regression model linking climate / land-use changes and species traits to short-term species trends. a** Schematic model representation. The response variable was 5-year species trends of regional mean occupancy ($n_{tot}$ = 20,048). Explanatory variables included changes in climate (annual mean temperature, temperature seasonality, summer precipitation) and land use (total agricultural area, grassland-use intensity, crop-use intensity), two trait variables, elevation (low or high) and interactions (indicated with ×) (cf. Fig. 1). In the restricted model version underlying the results presented in panel (**b**), parameter estimates for change in total agricultural area and grassland-use intensity (in italic) were only based on species of agriculturally influenced habitats (183 butterfly species, 93 grasshopper species; $n$ = 13,968). Non-independence within insect groups, time intervals, bioclimatic zones and species was accounted for. **b** Model results along the same arrangement as in (**a**). Curves show posterior distributions of model estimates; fill colours indicating effect sizes (positive values in blue, negative values red); dashed vertical lines indicate zero; numbers are posterior distribution means. Two-way interactions of change and trait variables with elevation are included such that upwards-facing curves show model estimates for high elevation and downwards-facing curves for low elevation (too low amounts of crop fields at high elevation to include interaction with crop-use intensity). Overlapping areas show other two-way interactions. The bottom centre panel shows how to interpret effect sizes. Starting at the overall mean occupancy, expected mean occupancy changes in 5 years when an explanatory variable is increased by one standard deviation are shown. All explanatory variables were standardised prior to analysis.

## Methods
All statistical analyses were performed through R version 4.1.0[50]. Besides the explicitly mentioned packages, the R packages cowplot[51], data.table[52], dplyr[53], ggplot2[54], itsadug[55], purrr[56], raster[57], sf[58], sfheaders[59], tibble[60] and tidyr[61] were key for data handling, data analysis, and plotting. Posterior distributions were summarised through means and credible intervals (CIs). CIs are the highest density intervals,

calculated through the package bayestestR[62]. To summarise multiple posterior distributions, 5000 Monte Carlo simulations were used.

### Study region
The study included data from the whole of Switzerland. As an observation unit for records, we chose 1 × 1 km squares (henceforth squares). Switzerland covers 41,285 km², spanning a large gradient in

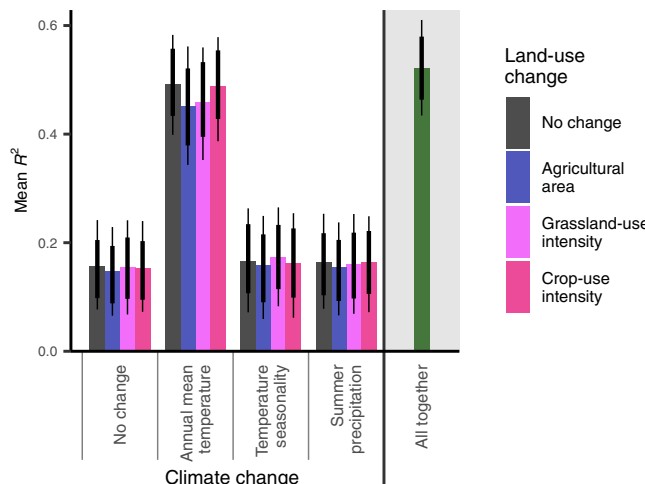

**Fig. 4 | Explained variance of long-term species trends for different climate and land-use scenarios.** Regression model results (Fig. 3) just reflect relationships of short-term trends to short-term changes, which can be of different relevance to explain long-term trends depending on the long-term change of drivers. Thus, based on the results of the regression model only including species of agriculturally influenced habitats (Supplementary Table S3), 40-year trends in mean occupancy (across the whole of Switzerland) were predicted for all 276 species for scenarios of no climate and land-use change (i.e., change assumed to be zero across all zones and time intervals) and for different combinations of single measured trajectories of climate and land-use variables. The $R^2$ values (squared Pearson correlations) indicate how well the predictions align with the observed long-term species trends. The green bar on the right shows the match for the predictions with all climate and land-use variables following their measured trajectories (analogous to the $R^2$ of the model presented in Fig. 3a). Bars show means of the posterior distribution; vertical lines show 80%- and 95% credible intervals.

elevation, climate and land use. It can be divided into five coarse biogeographic regions based on floristic and faunistic distributions and on institutional borders of municipalities[63] (Fig. 1b). The Jura is a mountainous but agricultural landscape in the northwest (~4200 km², 300–1600 m asl; annual mean temperature: ~9.4 °C, annual precipitation: ~1100 mm (gridded climate data here and in the following from MeteoSwiss (https://www.meteoswiss.admin.ch), average 1980–2020, at sites ~500 m asl.)). The Jura is separated from the Alps by the Plateau, which is the lowland region spanning from the southwest to the northeast (~11,300 km², 250–1400 m asl, mostly below 1000 m asl; ~9.5 °C, ~1100 mm). It is the most densely populated region with most intensive agricultural use. For the Alps, three regions can be distinguished. The Northern Alps (~10,700 km², 350–4000 m asl; ~9.2 °C, ~1400 mm), which entail the area from the lower Prealps, which are rather densely populated and largely used agriculturally, up to the northern alpine mountain range. The Central Alps (~11,300 km², 450–4600 m asl; ~9.5 °C, ~800 mm) comprise of the highest mountain ranges in Switzerland and the inner alpine valleys characterised by more continental climate (i.e., lower precipitation). Intensive agricultural land use is concentrated in the lower elevations and agriculture in higher elevations is mostly restricted to grassland areas used for summer grazing. The Southern Alps (~3800 km², 200–3800 m asl; ~10.4 °C, 1700 mm) range from the southern alpine mountain range down to the lowest elevations of Switzerland and are clearly distinguished from the other regions climatically, as they are influenced by Mediterranean climate, resulting in, e.g., milder winters. Besides differences between biogeographic regions, climate, land use and changes therein vary greatly between different elevations (Supplementary Fig. S9). To account for these differences, we split the regions in two elevation classes at the level of squares. All squares with a mean elevation of less than 1000 m asl were assigned to the low elevation, whereas squares above 1000 m asl were assigned to the high elevation

(no squares in the Plateau fell in the high elevation). This resulted in nine bioclimatic zones (Fig. 1b), for which separate species trends were estimated in the subsequent analyses. The threshold of 1000 m asl enabled a meaningful distinction based on the studied drivers (climate and land-use change) and was also determined by the availability of records data (high coverage in all nine bioclimatic zones).

## Species detection data
We extracted records of butterflies (refers here to Papilionoidea as well as Zygaenidae moths), grasshoppers (refers here to all Orthoptera) and dragonflies (refers here to all Odonata) from the database curated by info fauna (The Swiss Faunistic Records Centre; metadata available from the GBIF database at https://doi.org/10.15468/atyl1j, https://doi.org/10.15468/bcthst, https://doi.org/10.15468/fcxtjg). This database unites faunistic records made in Switzerland from various sources including both records by private persons and from projects such as research projects, Red-List inventories or checks of revitalisation measures. Only records with a sufficient precision, both temporally (day of recording) and spatially (place of recording known to the precision of 1 km² or less), were used for analyses. Besides temporal and spatial information, information on the observer and the project (if any) was obtained for each record. All records made by a person/ project on a day in a square were attributed to one visit, which was later used as replication unit to model the observation process (see below).

We included records from the focal time range 1980–2020. Additionally, we included records from 1970–1979 for butterflies in occupancy-detection models to increase the robustness of mean occupancy estimates. We excluded the mean occupancy estimates for these additional years from further analyses to cover the same period for all groups. Prior to analyses, following the approach in ref. 26, we excluded observations of non-adult stages and observations from squares that only were visited in 1 year of the studied period, because these would not contain any information on change between years[64]. This resulted in 18,018 squares (15,248 for butterflies, 9870 for grasshoppers, 5188 for dragonflies) and 1,448,134 records (879,207 butterflies, 272,863 grasshoppers, 296,064 dragonflies) that we included in the analyses (Supplementary Fig. S2). The three datasets for the different groups were treated separately for occupancy-detection modelling, following the same procedures for all three groups. To determine detections and non-detections for each species and visit, which could then be used for occupancy-detection modelling, we only included visits that (a) did not originate from a project, which had a restricted taxonomic focus not including the focal species, (b) were not below the 5% quantile or above the 95% quantile of the day of the year at which the focal species has been recorded[26] and (c) were from a bioclimatic zone, from which the focal species was recorded at least once.

## Occupancy-detection models
We used occupancy-detection models[65,66] to estimate annual mean occupancy of squares for the whole of Switzerland and for the nine bioclimatic zones for each species (i.e., mean number of squares occupied by a species), mostly following the approach in ref. 26. We fitted a separate model for each species, based on different datasets for the three groups. We included only species that were recorded in any square in at least 25% of all analysed years. Occupancy-detection models are hierarchical models in which two interconnected processes are modelled jointly, one of which describes occurrence probability (ecological process; used to infer mean occupancy), whereas the other describes detection probability (observation process)[65]. The two processes are modelled through logistic regression models. The occupancy model estimates occurrence probability for all square and year combinations, whereas the observation model estimates the probability that a species has been detected by an observer during a visit. More formally, each square $i$ in the year $t$ has the latent occupancy

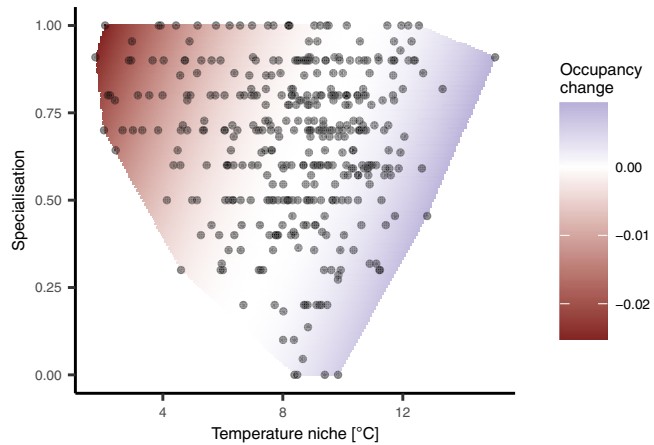

**Fig. 5 | Forty-year trends dependent on species traits.** Based on the regression model on 5-year trends (Fig. 3), average predicted change in occupancy within a bioclimatic zone across the whole 40-year study period (1980–2020) is shown in colour across the temperature niche as well as the habitat specialisation gradient of the 390 study species (points show the trait range of all species). In relation to mean 40-year occupancy across all species, a change in the occupancy of 0.01 corresponds to a decrease or an increase of the distribution of a species in a bioclimatic zone of 4.5%.

status $z_{i,t}$, which may be either 1 (present) or 0 (absent). $z_{i,t}$ depends on the occurrence probability $\psi_{i,t}$ as follows

$$z_{i,t} \sim \text{Bern}(\psi_{i,t}) \tag{1}$$

The occupancy status is linked to the detection/non-detection data $y_{i,t,j}$ at square $i$ in year $t$ at visit $j$ as

$$y_{i,t,j}|z_{i,t} \sim \text{Bern}(z_{i,t}p_{i,t,j}) \tag{2}$$

where $p_{i,t,j}$ is the detection probability.

The regression model for occurrence probability (occupancy model) looked as follows

$$\text{logit}(\psi_{i,t}) = \mu_o + \beta_{o1}\text{elevation}_i + \beta_{o2}\text{elevation}_i^2 + \alpha_{o1,i} + \alpha_{o2,i} + \gamma_{r(i),t} \tag{3}$$

with $\mu_o$ being the global intercept, elevation$_i$ being the scaled elevation above sea level and $\alpha_{o1,i}$, $\alpha_{o2,i}$ and $\gamma_{r(i),t}$ being the random effects for fine biogeographic region (12 levels, Supplementary Fig. S10; these were again defined based on floristic and faunistic distributions and followed institutional borders[63]), square and year. The random effects for fine biogeographic region and square were modelled as follows:

$$\alpha_{o1} \sim \text{Normal}(0,\sigma_{o1}) \tag{4}$$

and

$$\alpha_{o2} \sim \text{Normal}(0,\sigma_{o2}) \tag{5}$$

The random effect of the year was implemented with separate random walks per zone following ref. 67, which allowed the effect to vary between the nine bioclimatic zones, while accounting for dependencies among consecutive years. Conceptually, in random walks, the effect of 1 year is dependent on the previous year's effect, resulting in trajectories with less sudden changes between consecutive

years. This was implemented as follows:

$$\gamma_{r,t} \sim \begin{cases} \text{Normal}\left(0,1.5^2\right) & \text{for } t=1 \\ \text{Normal}\left(\gamma_{r,t-1},\sigma_{\gamma r}^2\right) & \text{for } t>1 \end{cases} \tag{6}$$

with

$$\sigma_{\gamma r} \sim \text{Cauchy}(0,1) \tag{7}$$

The regression model for detection probability (observation model) looked as follows

$$\begin{aligned}\text{logit}(p_{i,t,j}) = {}& \mu_d + \beta_{d1}\text{yday}_j + \beta_{d2}\text{yday}_j^2 + \beta_{d3}\text{shortlist}_j + \beta_{d4}\text{longlist}_j \\ & + \beta_{d5}\text{expert}_j + \beta_{d6}\text{project}_j + \beta_{d7}\text{targeted\_project}_j \\ & + \beta_{d8}\text{redlist}_j + \alpha_{d1,t} \end{aligned} \tag{8}$$

where $\mu_d$ is the global intercept, yday$_j$ is the scaled day of the year of visit $j$, shortlist$_j$ and longlist$_j$ are dummies of a three-level factor denoting the number of species recorded during the visit (1; 2–3; >3), and expert$_j$, project$_j$, targeted_project$_j$ and redlist$_j$ are dummies of a five-level factor denoting the source of the data. The source might either be a common naturalist observation (reference level), an observation by an expert naturalist, an observation made during a not further specified project, an observation made in a project targeted at the focal species or an observation made during a Red-List inventory. An expert naturalist was defined as an observer that contributed a significant number of records, which was defined as the upper 2.5% quantile of all observers arranged by their total number of records, and that made at least one visit with an exceptionally long species list, which was defined as a visit in the upper 2.5% quantile of all visits arranged by the number of records. The proportions of records originating from these different sources changed across years, but change was not unidirectional and differed among the investigated groups (Supplementary Fig. S11), such that accounting for data source in the model should suffice to yield reliable estimates of occupancy trends. $\alpha_{d1,t}$ is a random effect for year, which was modelled as

$$\alpha_{d1} \sim \text{Normal}(0,\sigma_{d1}) \tag{9}$$

The occupancy and observation models were fitted jointly in Stan through the interface CmdStanR[68]. Four Markov chain Monte Carlo chains with 2000 iterations each, including 1000 warm-up iterations, were used. Priors of the model parameters are specified in Supplementary Table S5. For the prior distribution of global intercepts, a standard deviation of 1.5 was chosen to not overweight the extreme values on the probability scale. To ensure that chains mixed well, Rhat statistics for annual mean occupancy estimates were calculated through the package rstan[69]. For Switzerland-wide annual estimates ($n = 18,140$), 98.0% of values met the standard threshold of 1.1 (99.9% of values <1.55 and all values <1.56). For the annual estimates per bioclimatic zone ($n = 116,844$), 98.3% of values met the 1.1 threshold (99.9% of values <1.55 and all values <1.62). To check the validity of the model results, we compared species trends estimated for butterflies to trends estimated from standardised samplings for the same region in a national monitoring programme and found them to be clearly correlated (Supplementary Fig. S3).

From the models of all three insect groups, we extracted the posterior distribution of the predicted annual mean occupancy per bioclimatic zone and for the whole of Switzerland for all 390 species. We used linear models to quantify species trends of mean occupancy against years for each draw of the posterior distribution. On the one hand, we determined global species trends in mean occupancy across

the whole of Switzerland for the time range 1980–2020. On the other hand, we determined short-term species trends in mean occupancy per bioclimatic zone for all consecutive 5-year intervals (8 intervals starting with 1980–1985). This was necessary to being able to analyse species trends against a set of uncorrelated climate and land-use change variables (see below). The length of the intervals was chosen to be ecologically meaningful while representing the variability in species trends adequately. Longer intervals might miss relations because short-term variability in drivers and trends is flattened out and analyses would lack replication of climate variables. Still, the main findings could be confirmed in additional analyses with 10-year intervals (Supplementary Table S6 and Supplementary Fig. S12).

### Climate and land-use change

We selected climatic variables from the commonly used set of 19 bioclimatic variables[70] to represent climate change. Selection was restricted to variables that are potentially most meaningful to explain insect trends[71] and not strongly correlated, resulting in three variables: annual mean temperature (BIO1) representing changes in absolute temperature; temperature seasonality (BIO4, defined as the standard deviation of monthly temperature means), representing changes in annual temperature cycles; and precipitation of the warmest quarter (BIO18) representing changes in precipitation during summer months (thus termed summer precipitation throughout). We determined yearly values for these variables for the whole of Switzerland from high-resolution mean monthly temperature and total monthly precipitation values reported by MeteoSwiss (https://www.meteoswiss.admin.ch) at a 1.25-degree minute grid (~2.3 × 1.6 km). Then, we calculated yearly means per bioclimatic zone while only considering grid cells intersecting with 1 × 1 km squares included in any of the occupancy-detection models. To infer climate change, we used linear models to determine trends in the three variables across consecutive 5-year intervals (same as for short-term species trends). To account for lags in effects of climate change[72] and to prevent large biases in trend estimates due to single extreme years, we also included the 5 years preceding each 5-year interval in the linear models for climatic variables (spanning 10 years in total, first interval 1975–1985).

We derived land-use variables at regional scale from the national agricultural statistics and censuses that were recorded by the Federal Statistical Office for the time range 1955–2020, from which we used data on total agricultural area, grassland area, livestock numbers and area of different crops (arable and permanent). These data were available at the level of 2172 municipalities in yearly resolution (1996–2020) or 5- to 10-year resolution for earlier years (Supplementary Table S7). The cover of other land-use types, such as forests, might be important for the studied species, but data were not available at the necessary temporal resolution. As additional analyses of available data of other land-use types indicated that changes of the most relevant land-use types (forests, built-up areas) were strongly related to changes in agricultural area (Supplementary Fig. S8), inclusion of agricultural area in our analyses sufficiently covered the main land-use changes in the last 40 years in the study area. We summarised livestock numbers into number of livestock units (LSU) based on commonly used weighting factors[73]. To aggregate data at the level of the nine bioclimatic zones, we distributed municipality-level data to agricultural land within municipalities based on the spatial distribution of agricultural areas within the municipalities[74]. Note that data from the national censuses are attributed to the municipality of the farm of the landowner, resulting in some degree of misclassification, which is, however, expected to level out at the aggregation level used for later analyses. We then aggregated the resulting spatially explicit data to the bioclimatic zones, considering only squares that were included in any of the occupancy-detection models to not overrepresent land use in largely unvisited regions (mainly high alpine regions). Because all included variables have little year-to-year variation, but rather change

gradually, we used generalised additive models to fill data gaps and to predict yearly values for the time range of interest (1980–2020) for all variables and bioclimatic zones (function stan_gamm4 in the package rstanarm[75], four Markov chain Monte Carlo chains with 2000 iterations each, including 1000 warm-up iterations). Based on these data, we derived three variables of land-use change at the level of bioclimatic zones. First, we determined the proportion of agricultural area as the proportion of total agricultural land in relation to the total study area. Second, we quantified grassland-use intensity as the mean number of LSU per grassland area. As such, it represents a general index of grazing, mowing and fertilisation pressure on the available grasslands. To check the validity of this measure, we compared trajectories of regional grassland-use intensity to grassland-use intensity inferred from satellite imagery, which showed very similar regional trends (Supplementary Fig. S13). Third, we determined crop-use intensity by attributing mean insecticide application rates to the most common crop types (including orchards and vineyards) (following ref. 76). Using these values, we calculated insecticide application rates for the total cropped area in a bioclimatic zone and aggregated it to an overall application rate per zone. To represent a mean insecticide pressure in the landscapes, we set the aggregated rates in proportion to the total study area per zone (note that here general additive models were only used on the aggregated application rate and not on the single crop types). Finally, we used changes across the investigated 5-year intervals as variables of land-use change.

### Species traits

Various traits have been linked to species' susceptibility to global change[23,31,49,77–81]. Here, we included two traits in our analyses of species trends that were expected to be strongly sensitive to the investigated drivers (climate and land-use changes). First, the temperature niche of each species may determine its response to climate change. We estimated the temperature niche following the Species Temperature Index approach[82] based on distributional records from the GBIF database (GBIF.org, https://doi.org/10.15468/dl.t6ha3h for butterflies, https://doi.org/10.15468/dl.reemkv for grasshoppers, https://doi.org/10.15468/dl.czbrmq for dragonflies). To reduce sampling biases, we only considered records from Europe and aggregated them at the Common European Chorological Grid Reference System (CGRS) grid. We extracted mean temperature (1970–2000) from WorldClim 2[83] at a 2.5 min spatial resolution and aggregated it at the CGRS grid. For each species, we determined the temperature niche as the mean temperature of the grid cells where it was recorded at least once. Second, we quantified habitat specialisation for all species, as specialist species, e.g., in terms of habitat or feeding preference, have repeatedly been shown to respond particularly strongly to land-use change and intensification[77,79,80]. We estimated habitat specialisation based on a database of ecological preferences[84], which compiles information on habitat preference specific for Switzerland for all studied species based on available literature and expert knowledge. We extended the database to include Zygaenidae based on relevant literature[85,86] and expert knowledge. For each species, the preferred habitats are given from a list of typical habitats, which are defined separately for each of the studied insect groups based on their habitat ranges (Supplementary Table S8). For grasshoppers, adult habitat preference is specified, whereas larval habitat preference is specified for butterflies and dragonflies. We defined a continuous habitat specialisation index $I_{\text{spec},i}$ for species $i$ as

$$I_{\text{spec},i} = 1 - \frac{n_i}{N_{\text{group}(i)}} \qquad (10)$$

where $n_i$ is the number of preferred habitats of species $i$ and $N_{\text{group}(i)}$ is the total number of habitats listed for the insect group. To obtain a global measure for habitat specialisation for each single species, we

standardised this index to values between 0 (least specialised) and 1 (most specialised) within insect groups.

## Analysis of species trends

We analysed trends in species mean occupancy at 5-year intervals in a regression model with changes in climate and land use (at the same time intervals) as well as species traits as explanatory variables (Fig. 3a). Besides the main effects of the three climate change variables (annual mean temperature, temperature seasonality, summer precipitation), the three land-use change variables (total agricultural area, grassland-use intensity, crop-use intensity), the species traits (temperature niche, habitat specialisation) and the elevation (low or high), we included a set of interactions, which were expected to affect species trends (Fig. 3a). First, because absolute values of climate and land-use variables differ most considerably between the two elevations (Supplementary Fig. S9), we included interactive effects between the elevation and climate as well as land-use change variables. Because crop-use intensity in high elevations was minimal due to generally low amounts of cropped area, we did not include the interactive effect between elevation and crop-use intensity. Second, following the same line of reasoning as for global change variables, we included interactive effects of species traits and elevation. Third, we included all interactive effects of climate and land-use change variables to test for synergistic or antagonistic effects between the two global change drivers. Fourth, because the temperature niche of a species is expected to affect its reaction to climate change, we included the interactive effects between temperature niche and the three climate change variables. Finally, because specialised species have been shown to be particularly susceptible to land-use change and intensification, we included the interactive effects between habitat specialisation and the three land-use change variables. In addition, we included separate intercepts for the three groups to account for potential differences in mean trends and a factorial covariate for the 5-year time interval to account for variance in trends not covered by the climate and land-use change variables. Also, we included the bioclimatic zone and species identity as random effects. For species identity, we not only included random intercepts, but also random slopes in respect to the global change drivers and their interactions, because different species were expected to react differently to these drivers. To meet the normality assumption for the residual distribution, we transformed species trends by taking the square root of their absolute values, while keeping their original sign. We scaled all continuous variables to standard deviation 1 and centred to mean 0 at the level of their recording prior to analyses. We fitted the model in Stan through the R interface rstan[69] (four Markov chain Monte Carlo chains with 2000 iterations each, including 1000 warm-up iterations; priors in Supplementary Table S5). To assess the model fit, we did posterior predictive model checking of residual distribution and checked for temporal autocorrelation in the residuals (Supplementary Fig. S14).

Not all study species are equally associated to agriculturally influenced habitats and might thus not be equally sensitive to agricultural land-use change. While crop-use intensity is expected to affect all habitats in a region due to the spread of insecticides to neighbouring terrestrial habitats[87] or water bodies[88], the effect of the other land-use change variables (total agricultural area, grassland-use intensity) on habitats that are not agriculturally influenced is less clear. As a result, we used three different versions of the above model to account for a potential bias. In the first version of the model, we made sure that all parameters that included the respective land-use change variables (total agricultural area, grassland-use intensity) were estimated based on observations from a subset of species, namely species that are at least partly associated to agriculturally influenced habitats. All other parameter estimates, however, were based on observations from the full set of species. The subset of species was defined based on the database of ecological preferences that was also

used to quantify habitat specialisation[84]. From the list of habitats, we defined agriculturally influenced habitats (Supplementary Table S8) and included only species that occur in at least one of them in the species subset. It contained 183 butterfly species (out of 215), 93 grasshopper species (out of 103) and no dragonfly species. To avoid underestimating effects of land-use changes due to the inclusion of species not directly associated to agriculturally influenced habitats when comparing climate and land-use effects (see scenario predictions below), the second version of the model did only include the species associated to agriculturally influenced habitats for all parameter estimates, i.e., was a full model run on a subset of the data. However, changes in an agricultural area or grassland-use intensity might also affect the species not directly associated to agriculturally influenced habitats. For example, a change in agricultural area indirectly also means a change in other habitats such as forests (Supplementary Fig. S8) and increasing grassland-use intensity might result in higher nutrient loads of adjacent water bodies[89]. Thus, the third version of the model was a full model with all species, where the full set of species was included for all parameter estimates. The findings based on the different model versions were largely consistent. We report parameter estimates from the first version in the main manuscript and results from the other versions in Supplementary Tables S3 and S4.

Additional sensitivity analyses were done based on the first model version. Because the data-based species selection also included some species for which mean occupancy estimates might be biased due to different reasons (migratory or (re)introduced species and species with uncertain taxonomic status or very difficult identification; Supplementary Table S1), we did sensitivity analyses, in which we excluded these species from the trend analyses. At the same time, mean occupancy estimates might be less reliable for species with only few records. Thus, we also excluded the species with the lowest record numbers (lowest 20% of species per group) for these sensitivity analyses. We found model results to be robust to these exclusions (Supplementary Table S9). Furthermore, mean occupancy estimates for some species in some bioclimatic zones might be unreliable because of low record numbers in these zones. In additional sensitivity analyses, we thus excluded species-zone combinations with very few records (<41, i.e., on average less than 1 record per year and zone) and again found model results to be robust (Supplementary Table S10).

To understand how the relations of short-term species trends to global change drivers explained the observed changes in species' mean occupancy across the 40 studied years (long-term trends), we predicted 40-year species trends for different climate and land-use change scenarios from the parameter estimates of the regression model. To prevent an underestimation of the land-use change effects, we used the second model version (i.e., only including species associated to agriculturally influenced habitats) for these model predictions. First, the basic scenario was composed as such that, starting from the original dataset, all climate and land-use change variables were held constant at their absolute 0 (uncentred variables), representing no change. All other variables (elevation, time interval, bioclimatic zone, temperature niche, habitat specialisation) were kept unchanged. Second, we defined additional scenarios on top of the basic scenario such that always one climate or land-use change variable was following its measured trajectory. Third, to account for interactive effects between climate and land-use change, we defined scenarios in which always two variables (one climate change, one land-use change) followed their measured trajectories. Finally, we defined a scenario in which all variables followed their measured trajectories. From all scenarios, we made predictions at the replication unit of the original dataset (one prediction per species and bioclimatic zone and time interval). We then back-transformed these short-term trend predictions (reversed root transformation) to represent mean occupancy changes across a 5-year interval, summed them per species and bioclimatic zone to represent 40-year trends and finally averaged long-

term trends per species while accounting for the number of squares per bioclimatic zone to yield an estimate of the Switzerland-wide mean occupancy change across 40 years for each species. We applied the same procedure to the observed short-term trend estimates. To evaluate how well the different scenarios reflected the observed trends, we determined $R^2$ values (squared Pearson correlations) for the comparison of predicted and observed long-term trends as a measure of explained variance for all scenarios.

## Data availability

The raw species records data are protected by a code of conduct common to all Swiss national data centres but might be obtained from info fauna upon request if in accordance with the code of conduct. Species records data at a coarser spatial resolution are available from the GBIF database (https://doi.org/10.15468/atyl1j, https://doi.org/10.15468/bcthst, https://doi.org/10.15468/fcxtjg). The regional annual mean occupancy data generated in this study have been deposited in the envidat database[90] (https://doi.org/10.16904/envidat.355). The climate and land-use change data are under restricted access but might be directly obtained from the sources mentioned in the Methods. The GBIF records data used in this study are available from the GBIF database (https://doi.org/10.15468/dl.t6ha3h, https://doi.org/10.15468/dl.reemkv, https://doi.org/10.15468/dl.czbrmq). The WorldClim 2 data used in this study are available from the WorldClim database (https://www.worldclim.org). The species' ecological preferences data used in this study are available from the Fauna Indicativa database (http://www.cscf.ch/cscf/de/home/projekte/fauna-indicativa.html). The species traits data generated in this study are provided in the Supplementary Information. The spatial data (biogeographic regions, elevation, municipalities) for Switzerland used in this study are available from the geoportal of the Swiss Confederation (https://data.geo.admin.ch; ch.bafu.biogeographische_regionen, ch.swisstopo.swissalti3d, ch.swisstopo.swissboundaries3d-gemeinde-flaeche.fill).

## Code availability

All relevant codes used to complete this work have been deposited in a GitHub repository (https://github.com/nefff1/insect_trends) available from Zenodo[91] (https://doi.org/10.5281/zenodo.7318603).

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

## Acknowledgements

We are greatly thankful to all project managers, experts, and naturalist, who provided species data to the info fauna database. We thank A. Berger, M. Bencheikh-Latmanj, A. Indermaur, G. Litsios, M. Obrist, S. Peter and R. Siber for valuable inputs or help in data provision.

## Author contributions

F.K.N., E.R., M.A., K.B., F.C., M.M.G., M.M., F.H. and E.K. initiated the project; F.N., F.K.N., E.R. and E.K. conceived the idea for the manuscript and defined the final analyses; F.N., E.R., Y.C., E.S.M., C.M. and T.R. collected and processed data; F.N. analysed the data; F.N. wrote the first manuscript draft, with input from F.K.N., E.R. and E.K.; F.N., F.K.N., E.R., M.A., K.B., F.C., Y.C., M.M.G., C.M.N., E.S.M., C.M., M.M., T.R., F.H. and E.K. discussed the analyses and commented on the manuscript.

## Competing interests

The authors declare no competing interests.

## Additional information

[1]Agroecology and Environment, Agroscope, Reckenholzstrasse 191, 8046 Zürich, Switzerland. [2]Swiss Ornithological Institute, Seerose 1, 6204 Sempach, Switzerland. [3]info fauna, Avenue de Bellevaux 51, 2000 Neuchâtel, Switzerland. [4]Biodiversity and Conservation Biology, Swiss Federal Research Institute WSL, Zürcherstrasse 111, 8903 Birmensdorf, Switzerland. [5]Department of Crop Sciences, Research Institute of Organic Agriculture FiBL, Ackerstrasse 113, 5070 Frick, Switzerland. [6]Forest Entomology, Swiss Federal Research Institute WSL, Zürcherstrasse 111, 8903 Birmensdorf, Switzerland. [7]Department of Environmental Systems Science, Institute of Terrestrial Ecosystems, ETH Zurich, Universitätstrasse 16, 8092 Zürich, Switzerland. [8]Department of Environmental Sciences, Zoology, University of Basel, Vesalgasse 1, 4051 Basel, Switzerland. [9]Hintermann & Weber AG, Austrasse 2a, 4153 Reinach, Switzerland. [10]Department of Evolutionary Biology and Environmental Studies, University of Zurich, Winterthurerstrasse 190, 8057 Zürich, Switzerland. ✉e-mail: mail@felixneff.ch

