## [Peer Review File · Nature Communications]

REVIEWERS' COMMENTS

Reviewer #1 (Remarks to the Author):

In general, I found this revised draft of the paper a substantial improvement over the original (reviewed for Nature). The authors are now consistent in specifying that their comparison is between impacts of climate change and of REGIONAL land use change, rather than of land use change in general, and they have added a substantial paragraph (lines 169-182) explaining the limitations of the approach. Nonetheless, the title and abstract give the clear impression that (overall) land use impacts are small relative to those of climate, and so I fear that many readers may not notice the subtleties of the paper as a whole. In my mind, the strongest findings here are not only the direct impacts of climate change, but the degree to which it interacts with land use change and with species' traits. The climate land-use interaction is largely missing from the title and abstract (although the species' trait effect is mentioned in the abstract). Conversely, the abstract covers the properties of the increasing and decreasing quartiles of species in considerable detail -- which seemed an odd thing to emphasize (especially as the metric is directionally biased: a species cannot lose more than 100% of its occupancy, but it can gain substantially more than 100%, especially if its occupancy is low; perhaps a logit transform would be appropriate?).

Beyond that, there were some relatively minor points:

Line 52 "as well as in politics" (awkward)

Line 66 "barely"? Perhaps "seldom"?

Lines 74 (and 259-260): make it clear that all Orthoptera and Odonata are included, not just grasshoppers and dragonflies.

Lines 97-99: Fig S3 provides some support for your models, but I'm not sure that it's fair to describe them as "well aligned"?

Line 129: I'm not sure that grasshoppers and dragonflies contained "more" warm adapted species in absolute terms, as they had fewer spp altogether (103 grasshoppers and 72 dragonflies, vs. 215 butterflies); a greater proportion perhaps?

Line 193: "decreased on the expense of scrubs and forests"? Hard to understand. Do they mean agricultural land decreases were linked to increased cover of scrub & forest?

Reviewer #2 (Remarks to the Author):

For the record, I am not connected in any way to the references or resources cited below. I was reviewer 2 for the paper submitted to Nature, that has now been passed to NCOMMS.

Thank you for addressing my questions related to random effects, random walks and temporal autocorrelation. Thank you also for adding the absolute changes in lines 101 onwards, but this insertion in the main body of the text ignores the fact that the abstract is much more sensationalist. For the absolute change in occupancy for declining species, the number seems very slight (0.0660). The authors write in lines 106 onwards "For the 25% species with the strongest negative trends (declining quarter), mean occupancy, i.e. the proportion of total squares that are occupied, on average decreased across the 40 years by 0.0660 (95%-CI: 0.0613-0.0709)".

In the abstract, much is made of the strong declines without an absolute measure inserted and I am concerned about context. There, the authors write "For species that showed strongest declines, average mean occupancy decreased by 58.3% (95% credible interval: 52.2-64.4%) relative to their 40-year mean occupancy". I see this as hyping the results and would like to see absolute numbers in the abstract too to give real context. Further, on the first line of the discussion the authors state "Across the whole of Switzerland, there was no overall decline in species distributions, but strong decreases and increases of individual species". Again, the abstract should state that there is no overall decline, but it does not appear.

Drivers

Regarding the response to my comments on drivers and their relevance to the fauna studied (i.e. "I

am concerned by the limited landscape driver dataset - total agricultural area, grassland-use intensity, crop-use intensity. The paper makes no mention anywhere in the submission of woodland or forest habitats and for dragonflies in particular the presence and size of wetland habitats").

The authors kindly responded to this comment by stating that "We would also have liked to include more information on forest and wetland changes, but were not able to do so, as the data was not available at the same temporal and spatial resolution as the land-use data we were using."

I am still not convinced by the driver approach for dragonflies and the lack of any relevant habitat related analyses. If I were to start a new experiment on monitoring change in this group, I would first measure wetland area and water quality, and low on my list would be total agricultural area, grassland-use intensity, crop-use intensity. Whilst I see that the authors have gone as far as the available data allows, an analyses should not be included in a manuscript just because a model can compute parameters.

The statement on 355 that "As additional analyses of available data of other land-use types indicated that changes of the most relevant land-use types (forests, built-up areas) were strongly related to changes in agricultural area (Fig. S7), inclusion of agricultural area in our analyses sufficiently covered the main land-use changes in the last 40 years in the study area." This might well be true, but it does not come close to addressing the void in wetland ecology. Therefore, it's not a surprise to me that 40-year changes were not clearly related to regional land-use changes for this group. Odonates species trends increased which as the authors stated "reflect improvements of water quality and wetland habitats made in the last decades", but are not measured.

Please remove all land-use driver changes from the odonates. It's not credible. Also, caveat any woodland butterfly and grasshopper species stating that these habitats were not modelled using their specific resident habitat as a driver, which may distort results. I appreciate the result on line 355 that "The cover of other land-use types, such as forests, might be important for the studied species, but data were not available at the necessary temporal resolution." I ask, would much impact be lost if you removed the few woodland species from the analyses? Then I think you have a water-tight paper.

Minor

1. but barely available = 'rarely' is better
2. 171 Excensfication = extensification
3. 179 last 40 years, BUT we could not detect such relations given the regional scale of our approach. Add 'but'.
4. 192 At high elevation, agricultural land decreased 'on the expense of scrubs and forests' = 'at the expense of scrub and forests'

Reviewer #3 (Remarks to the Author):

Summary

The study assesses the responses of species occurrence trends to combinations of climatic and land use variables for species of butterfly, dragonfly and grasshopper in Switzerland for the time period 1980-2020.

Rather than using a space for time substitution, as recent studies have done, this work is temporal in nature. Trends over time are assessed over the long term, and then short-term trends are used to establish the relationships between species trends and climate and land use variables. The effect sizes from this model are then used to project the potential long-term trends. This shows that changes in temperature are likely the most important variable for explaining recent trends for these species.

Bayesian occupancy models are used to determine trends over time. Given the nature of the occurrence data used these are the most appropriate methods to account for the unstandardised nature of data collection. Then a regression model is used to assess the effect of the climate and land use variables, plus species level metrics of habitat and temperature requirements.

This is my second review of this work, and I can say that it is much improved. The roles of the long term and short-term trends are much clearer now, and the methods are more detailed. I think it is a great study that is adding to the literature unpicking the roles of land use and climate change effects and their interactions on insect biodiversity.

Most of my comments are rather minor at this point.

Abstract

Line 45 - It is not clear what is meant by "reflecting mixed changes". This doesn't quite fit with the previous about weak relationships with land use change.

Introduction

The introduction nicely sets things up and includes all recent relevant literature as far as I can tell.

Line 82- 84 - I still think the clarity could be improved here. I understand, from your response to previous comments, why the short-term trends were used (to ensure uncorrelated land use and temperature data, and to enable replication), but it is still not obvious from this one line. Here is a suggestion if it is helpful: in a separate analysis, we assessed trends for consecutive 5-year intervals. This allowed an analysis of climate and land use variables over the same time period which are otherwise highly correlated over longer time periods. This also meant that we had considerable replication across climate and land use variables when split across bioclimatic zones and years.

Line 86 - trends is used here to describe the short-term assessment, but it could be misleading. Suggest reworking this sentence slightly to ensure the trait section is related to the short-term trends.

Results and discussion

The main description of the long-term trends is clear, with a nice separation of increasing vs decreasing trends and common vs rare species.

It is now much clearer where the long-term and short-term trends are used and for which part of the analysis.

Figure S4 is a nice demonstration of the differences in response between species IUCN categories.

Line 171 -extensification?.

I like figure S9. It really clearly shows the differences in trends for the levels of specialism. I would have like that to be in the main text but understandable if short on space.

Methods:

Line 300 - repetition in sentence and brackets, suggest removing one or the other.

450 - there is no detail on the posterior predictive checking and the related figure presents just the autocorrelation check. Can information on these checks be provided and what the results of these showed? It can be useful to provide this for others following similar methods but new to the area.

Supplementary:

There are no line number so sorry if this gets confusing!

Page 2, second line beneath equation 2: refers to figure S5, but I think it should be S10.

Page 3, third paragraph starting "Priors": Refers to Table S6 but should be S3.

REVIEWERS' COMMENTS

We want to thank the three reviewers and the editor for their time and valuable suggestions that again clearly improved the manuscript. A detailed point-by-point reply to all reviewer comments is given below.

Reviewer #1 (Remarks to the Author):

In general, I found this revised draft of the paper a substantial improvement over the original (reviewed for Nature). The authors are now consistent in specifying that their comparison is between impacts of climate change and of REGIONAL land use change, rather than of land use change in general, and they have added a substantial paragraph (lines 169-182) explaining the limitations of the approach. Nonetheless, the title and abstract give the clear impression that (overall) land use impacts are small relative to those of climate, and so I fear that many readers may not notice the subtleties of the paper as a whole. In my mind, the strongest findings here are not only the direct impacts of climate change, but the degree to which it interacts with land use change and with species' traits. The climate land-use interaction is largely missing from the title and abstract (although the species' trait effect is mentioned in the abstract). Conversely, the abstract covers the properties of the increasing and decreasing quartiles of species in considerable detail -- which seemed an odd thing to emphasize (especially as the metric is directionally biased: a species cannot lose more than 100% of its occupancy, but it can gain substantially more than 100%, especially if its occupancy is low; perhaps a logit transform would be appropriate?).

Thank you for the positive evaluation of the revised version of the manuscript.

As suggested, we have changed the title such that it now states that climate and land-use effects were different, but readers need to consult the abstract for more details on these differences. We also highlight the important findings of interactive effects of climate and land use change in the abstract more prominently. The new title is intended to avoid conveying potentially misleading messages to "quick readers". It now reads: "Past 40 years' insect trends: The different roles of concurring climate and regional land-use changes".

The more prominent mentioning of the interactive effects in the abstract now reads: "Short-term distribution changes were associated to both climate changes and regional land-use changes. Moreover, interactive effects between climate and regional land-use changes confirm that the various drivers of global change can have even greater impacts on biodiversity in combination than alone." (L39-42).

The idea to give percentages instead of absolute numbers was to give an idea of the magnitude of change in community composition, which is considerable. We chose a metric that is relative to mean occupancy (i.e., change / mean occupancy), which can take the same range of values both for increases and decreases. We slightly reformulated the text in the abstract as well as in the introduction to make this clear. Based on comments by reviewer 2, we also include absolute changes in occupancy in the abstract. It now reads: "For species that showed strongest increases (25% quantile), the average proportion of occupied squares increased in 40 years by 0.128 (95% credible interval: 0.123–0.132), which equals an average increase in mean occupancy of 71.3% (95%-CI: 67.4–75.1%) relative to their 40-year mean occupancy. For species that showed strongest declines (25% quantile), the average proportion decreased by 0.0660 (95%-CI: 0.0613–0.0709), equalling an average decrease in mean occupancy of 58.3% (95%-CI: 52.2–64.4%)." (L32-38)

Beyond that, there were some relatively minor points:

Line 52 "as well as in politics" (awkward)

We changed the wording, it now reads: "Therefore, recent reports on their decline raised major concerns in the scientific community as well as among policy makers, stakeholders and the general public." (L49-50)

Line 66 "barely"? Perhaps "seldom"?

Thank you, changed as suggested.

Lines 74 (and 259-260): make it clear that all Orthoptera and Odonata are included, not just grasshoppers and dragonflies.

Thank you, we slightly changed to wording to make this clear, it now reads: "... butterflies (refers here to all Papilionoidea as well as to Zygaenidae moths), grasshoppers (refers here to all Orthoptera) and dragonflies (refers here to all Odonata) ..." (L71-73)

Lines 97-99: Fig S3 provides some support for your models, but I'm not sure that it's fair to describe them as "well aligned"?

We slightly changed the wording to not overstate the alignment in this comparison. It now reads: "For butterflies, data from a standardized 17-year monitoring were available, which showed trends that broadly aligned with the estimated mean occupancy trends, supporting the validity of our models." (L98-100)

Line 129: I'm not sure that grasshoppers and dragonflies contained "more" warm adapted species in absolute terms, as they had fewer spp altogether (103 grasshoppers and 72 dragonflies, vs. 215 butterflies); a greater proportion perhaps?

You are right, thank you for raising this point. We changed the wording of this sentence such that it now reflects a difference in their proportion and not in absolute numbers. It now reads: "... contain species that were on average more warm-adapted than butterfly species ..." (L137-138).

Line 193: "decreased on the expense of scrubs and forests"? Hard to understand. Do they mean agricultural land decreases were linked to increased cover of scrub & forest?

Yes, that is what we meant. We have rephrased the text which now reads: "... decreases in the cover of agricultural land were related to increases in the cover of scrubs and forests." (L206-207)

Reviewer #2 (Remarks to the Author):

For the record, I am not connected in any way to the references or resources cited below. I was reviewer 2 for the paper submitted to Nature, that has now been passed to NCOMMS.

Thank you for addressing my questions related to random effects, random walks and temporal autocorrelation. Thank you also for adding the absolute changes in lines 101 onwards, but this insertion in the main body of the text ignores the fact that the abstract is much more sensationalist. For the absolute change in occupancy for declining species, the number seems very slight (0.0660). The authors write in lines 106 onwards "For the 25% species with the strongest negative trends (declining quarter), mean occupancy, i.e. the proportion of total squares that are occupied, on average decreased across the 40 years by 0.0660 (95%-CI: 0.0613–0.0709)". In the abstract, much is made of the strong declines without an absolute measure inserted and I am concerned about context. There, the authors write "For species that showed strongest declines, average mean occupancy decreased by 58.3% (95% credible interval: 52.2–64.4%) relative to their 40-year mean occupancy". I see this as hyping the results and would like to see absolute numbers in the abstract too to give real context. Further, on the first line of the discussion the authors state "Across the whole of Switzerland, there was no overall decline in species distributions, but strong decreases and increases of individual species". Again, the abstract should state that there is no overall decline, but it does not appear.

We never intended to be sensationalist and we apologize if our wording was misleading. The raw number is just very hard to understand if one has no context. In fact, a change in occupancy of 0.0660 might not be that slight at all if mean occupancy is in the same range (which it is for many species). We only reported relative results, as these are much easier to grasp and give the necessary context. However, we now also provide the absolute numbers in the abstract, as suggested. It now reads: "For species that showed strongest increases (25% quantile), the average proportion of occupied squares increased in 40 years by 0.128 (95% credible interval: 0.123–0.132), which equals an average increase in mean occupancy of 71.3% (95%-CI: 67.4–75.1%) relative to their 40-year mean occupancy. For species that showed strongest declines (25% quantile), the average proportion decreased by 0.0660 (95%-CI: 0.0613–0.0709), equalling an average decrease in mean occupancy of 58.3% (95%-CI: 52.2–64.4%)." (L32-38). Furthermore, we now include a statement on the lack of a general decline as well. (L31)

Drivers

Regarding the response to my comments on drivers and their relevance to the fauna studied (i.e. "I am concerned by the limited landscape driver dataset - total agricultural area, grassland-use intensity, crop-use intensity. The paper makes no mention anywhere in the submission of woodland or forest habitats and for dragonflies in particular the presence and size of wetland habitats").

The authors kindly responded to this comment by stating that "We would also have liked to include more information on forest and wetland changes, but were not able to do so, as the data was not available at the same temporal and spatial resolution as the land-use data we were using."

I am still not convinced by the driver approach for dragonflies and the lack of any relevant habitat related analyses. If I were to start a new experiment on monitoring change in this group, I would first measure wetland area and water quality, and low on my list would be total agricultural area, grassland-use intensity, crop-use intensity. Whilst I see that the authors have gone as far as the available data allows, an analyses should not be included in a manuscript just because a model can compute parameters.

The statement on 355 that "As additional analyses of available data of other land-use types indicated that changes of the most relevant land-use types (forests, built-up areas) were strongly related to changes in agricultural area (Fig. S7), inclusion of agricultural area in our analyses sufficiently covered the main land-use changes in the last 40 years in the study area." This might well be true, but it does not come close to addressing the void in wetland ecology. Therefore, it's not a surprise to me that 40-year changes were not clearly related to regional land-use changes for this group. Odonates species trends increased which as the authors stated "reflect improvements of water quality and wetland habitats made in the last decades", but are not measured.

Please remove all land-use driver changes from the odonates. It's not credible. Also, caveat any woodland butterfly and grasshopper species stating that these habitats were not modelled using their specific resident

habitat as a driver, which may distort results. I appreciate the result on line 355 that "The cover of other land-use types, such as forests, might be important for the studied species, but data were not available at the necessary temporal resolution." I ask, would much impact be lost if you removed the few woodland species from the analyses? Then I think you have a water-tight paper.

Although we still think that including the full set of species is a sound approach given that also non-agricultural habitats such as water bodies are influenced by e.g. change in grassland-use intensity through nutrient influx / eutrophication, we acknowledge your point and follow your advice to perform analyses without those taxonomic groups and species that might not or less directly be affected by agricultural habitats and their land-use type and intensity. This was not a small change but required large-scale reanalyses. To not remove one of the three species groups (dragonflies) completely from the trend regression models, which are a crucial part of the study, we chose an approach in which we would only include a subset of species clearly associated to agriculturally influenced habitats in the estimation of the parameters for change in agricultural area and grassland-use intensity (i.e. exclusion of dragonflies and other species only associated to non-agriculturally influenced habitats such as forests as suggested). We did, however, include all species for parameter estimates of climate change variables, trait variables as well as crop-use intensity change as we expected all species to be affected by those variables. We did not only expect climate change to affect all species but also crop-use intensity change because crop-use intensity is quantified based on insecticide application rates in a region. Insect application rates are also highly relevant for other adjacent habitats because insecticides are known to spread to neighbouring habitats (e.g. Brühl et al. 2021, Scientific Reports) or to water bodies (e.g. Stehle et al. 2018, Science of the Total Environment). Besides this restricted modelling approach, we include in the appendix the results for full models (i) only based on the subset of species with associations to agriculturally influenced habitats and (ii) based on all species (i.e. results included in the previous version of the manuscript). We think all three approaches have their arguments and the large agreement and consistency between results from the different models further underpin the validity and robustness of our findings. A detailed description of the revised approach is given at L511-538. As we chose to include the restricted model parametrisation in the main manuscript, some small rewriting of the results/discussion section was necessary (e.g. L158-160, L198-207, L216).

Minor

1. but barely available = 'rarely' is better

Thank you, we changed that.

2. 171 Excensification = extensification

Thank you for spotting this typo.

3. 179 last 40 years, BUT we could not detect such relations given the regional scale of our approach. Add 'but'.

Changed as suggested.

4. 192 At high elevation, agricultural land decreased 'on the expense of scrubs and forests' = 'at the expense of scrub and forests'

Thank you, we changed the sentence based on a comment made by reviewer 1.

Reviewer #3 (Remarks to the Author):

Summary

The study assesses the responses of species occurrence trends to combinations of climatic and land use variables for species of butterfly, dragonfly and grasshopper in Switzerland for the time period 1980-2020.

Rather than using a space for time substitution, as recent studies have done, this work is temporal in nature. Trends over time are assessed over the long term, and then short-term trends are used to establish the relationships between species trends and climate and land use variables. The effect sizes from this model are then used to project the potential long-term trends. This shows that changes in temperature are likely the most important variable for explaining recent trends for these species.

Bayesian occupancy models are used to determine trends over time. Given the nature of the occurrence data used these are the most appropriate methods to account for the unstandardised nature of data collection. Then a regression model is used to assess the effect of the climate and land use variables, plus species level metrics of habitat and temperature requirements.

This is my second review of this work, and I can say that it is much improved. The roles of the long term and short-term trends are much clearer now, and the methods are more detailed. I think it is a great study that is adding to the literature unpicking the roles of land use and climate change effects and their interactions on insect biodiversity.

Thank you for the very positive evaluation of our revised manuscript.

Most of my comments are rather minor at this point.

Abstract

Line 45 - It is not clear what is meant by "reflecting mixed changes". This doesn't quite fit with the previous about weak relationships with land use change.

We clarify this by relating the mixed changes to local land use.

Introduction

The introduction nicely sets things up and includes all recent relevant literature as far as I can tell.

Thank you.

Line 82- 84 – I still think the clarity could be improved here. I understand, from your response to previous comments, why the short-term trends were used (to ensure uncorrelated land use and temperature data, and to enable replication), but it is still not obvious from this one line. Here is a suggestion if it is helpful: in a separate analysis, we assessed trends for consecutive 5-year intervals. This allowed an analysis of climate and land use variables over the same time period which are otherwise highly correlated over longer time periods. This also meant that we had considerable replication across climate and land use variables when split across bioclimatic zones and years.

Thank you for this suggestion, which we partially adapted. It now reads: " In addition to using independent distribution trends for the nine zones, we assessed trends for consecutive 5-year intervals. The aim of the choice of this spatial and temporal resolution was to get a sufficient number of replicates of trends under different climate and land-use conditions that showed enough variance while their covariance was low (Fig. S1). Also, we considered the 5-year intervals to be most meaningful for the studied insect groups from an ecological perspective. Nonetheless, we run the analyses based on 10-year intervals and confirmed the main findings." (L81-87)

Line 86 – trends is used here to describe the short-term assessment, but it could be misleading. Suggest reworking this sentence slightly to ensure the trait section is related to the short-term trends.

As we extrapolate these trait effects to long-term trends as well, we feel that the present statement suits well. We therefore would suggest not to change anything here.

Results and discussion

The main description of the long-term trends is clear, with a nice separation of increasing vs decreasing trends and common vs rare species.

Thank you.

It is now much clearer where the long-term and short-term trends are used and for which part of the analysis.

Thank you.

Figure S4 is a nice demonstration of the differences in response between species IUCN categories.

Thank you.

Line 171 –extensification?.

This is now corrected.

I like figure S9. It really clearly shows the differences in trends for the levels of specialism. I would have like that to be in the main text but understandable if short on space.

Thank you for this excellent suggestion. We have moved the figure to the main manuscript as proposed.

Methods:

Line 300 – repetition in sentence and brackets, suggest removing one or the other.

Thank you for spotting this mistake, which has been corrected now.

450 – there is no detail on the posterior predictive checking and the related figure presents just the autocorrelation check. Can information on these checks be provided and what the results of these showed? It can be useful to provide this for others following similar methods but new to the area.

These are just standard procedures for residual distribution checks also used in frequentist statistics. We now added more detailed information to this sentence. It now reads: "To assess the model fit, we did posterior predictive model checking of residual distribution and checked for temporal autocorrelation in the residuals" (L508-510).

Supplementary:

There are no line number so sorry if this gets confusing!

Page 2, second line beneath equation 2: refers to figure S5, but I think it should be S10.
Thank you for spotting this error, which has been corrected.

Page 3, third paragraph starting "Priors": Refers to Table S6 but should be S3.
Thank you, this error has been corrected as well.